# Sekai: A Video Dataset towards World Exploration

**Zhen Li**[1,2,4*§] **Chuanhao Li**[1*†], **Xiaofeng Mao**[1], **Shaoheng Lin**[1], **Ming Li**[1],
**Shitian Zhao**[1], **Zhaopan Xu**[1], **Xinyue Li**[1], **Yukang Feng**[3], **Jianwen Sun**[3],
**Zizhen Li**[3], **Fanrui Zhang**[3], **Jiaxin Ai**[3], **Zhixiang Wang**[5], **Yuwei Wu**[2,4†],
**Tong He**[1], **Jiangmiao Pang**[1], **Yu Qiao**[1], **Yunde Jia**[4], **Kaipeng Zhang**[1,3†‡]

[1]Shanghai AI Laboratory [2]Beijing Institute of Technology
[3]Shanghai Innovation Institute [4]Shenzhen MSU-BIT University
[5]The University of Tokyo

https://lixsp11.github.io/sekai-project/

## Abstract

Video generation techniques have made remarkable progress, promising to be the foundation of interactive world exploration. However, existing video generation datasets are not well-suited for world exploration training as they suffer from some limitations: limited locations, short duration, static scenes, and a lack of annotations about exploration and the world. In this paper, we introduce Sekai (meaning "world" in Japanese), a high-quality first-person view worldwide video dataset with rich annotations for world exploration. It consists of over 5,000 hours of walking or drone view (FPV and UVA) videos from over 100 countries and regions across 750 cities. We develop an efficient and effective toolbox to collect, pre-process and annotate videos with location, scene, weather, crowd density, captions, and camera trajectories. Comprehensive analyses and experiments demonstrate the dataset's scale, diversity, annotation quality, and effectiveness for training video generation models. We believe Sekai will benefit the area of video generation and world exploration, and motivate valuable applications.

## 1 Introduction

*Explore. Dream. Discover.* — *Mark Twain*

World exploration and interaction form the foundation of humankind's odyssey, which are practical scenarios for world generation models [1]. These models aim to adhere to the world laws (real world or games) while facilitating unrestricted exploration and interaction within environments. In this paper, we focus on the first act of world generation—world exploration, which aims to use image, text, or video to construct a dynamic and realistic world for interactive and unrestricted exploration.

Recent advancements in video generation [2, 3, 4, 5, 6] have been remarkable, making it a promising approach for world generation through video generation. Meanwhile, camera-controlled video generation [7, 8, 9] is a suitable way for world exploration, since camera trajectories can be converted by keyboard and mouse inputs. However, generating *long* and *realistic* videos with *precise* camera control remains a significant challenge. A major bottleneck lies in the data itself. Existing video

---

§This work was done during the internship at Shanghai AI Laboratory.
*Equal contribution.
†Corresponding authors: wuyuwei@bit.edu.cn; lichuanhao@pjlab.org.cn; zhangkaipeng@pjlab.org.cn
‡Project leader.

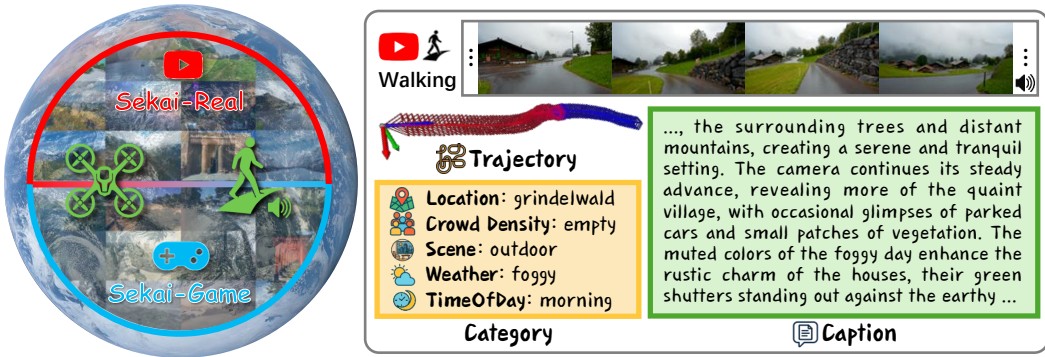

Figure 1: Sekai is collected from Youtube and a video game. It consists of walking and drone-view egocentric videos with recorded audio. We provide rich annotations of camera trajectories, location, crowd density, scene, weather, time of day, and captions.

generation datasets [10, 11, 12] are not well-suited for world exploration as they suffer from limitations: limited locations, short duration, static scenes, and a lack of annotations about exploration (*e.g.*, camera trajectories) and world annotations (*e.g.*, location, weather and scene).

In this paper, we introduce Sekai (せかい, meaning "world" in Japanese), a high-quality egocentric worldwide video dataset for world exploration (see Figure 1 and Figure 2). Most videos contain audio for an immersive world generation. It also benefits other applications, such as video understanding, navigation, and video-audio co-generation. Sekai-Real comprises over 5000 hours of videos collected from YouTube with high-quality annotations. Sekai-Game comprises videos from a realistic video game, Lushfoil Photography Sim, with ground-truth annotations. It has five distinct features: (1) **High-quality and diverse video**. All videos are recorded in 720p at 30 FPS, featuring diverse weather conditions, various times, and dynamic scenes. (2) **Worldwide location**. Videos are captured across 101 countries and regions, featuring over 750 cities with diverse cultures, activities, architectures, and landscapes. (3) **Walking and drone view**. Beyond the walking videos (*e.g.*, citywalk and hiking), Seikai contains drone view (FPV and UAV) videos for unrestricted world exploration. (4) **Long duration**. All walking videos are at least 60 seconds long, ensuring real-world, long-term world exploration. (5) **Rich annotations**. All videos are annotated with location, scene, weather, crowd density, captions, and camera trajectories. YouTube videos' annotations are of high quality, while annotations from the game are considered ground truth.

To construct the Sekai dataset, we develop a curation pipeline (see Section 3) for Sekai-Real (YouTube videos) and Sekai-Game (video game videos). (1) For Sekai-Real, we first manually search and download high-quality walking and drone videos. Then we introduce a pre-processing pipeline to obtain video clips by shot detection, video transcoding, and quality evaluation. After that, we develop an annotation framework to annotate location, scene type, weather, crowd density, captions, and camera trajectories. Considering the large amount of data and practical usage, we further introduce a video sampling module to sample the top-tier videos according to the computational resources for training the video generation model. (2) For the Sekai-Game, we first play Lushfoil Photography Sim and record videos. Then we use the same pre-processing pipeline to obtain video clips. For the annotation, we develop a toolbox to record ground-truth annotations while playing.

We conduct statistical analyses to characterize the scale and diversity of the dataset and independently validate the accuracy of YouTube annotations. We then fine-tune a video generation foundation model on the the top tier of Sekai-Real for text-to-video and image-to-video, yielding consistent gains in world-exploration scenarios, especially in video dynamics and visual quality. In addition, leveraging Sekai's camera trajectory annotations, we train for interactive video generation, where the model takes a camera trajectory as input and generates videos consistent with the intended camera motion. Across Sekai-Real and Sekai-Game, this training substantially improves interaction following, significantly reducing the error between the trajectories of the generated video and the target.

To summarize, our contributions are threefold:

- We introduce Sekai, a large-scale, high-quality long-form video dataset for worldwide exploration via walking and drone footage, with rich annotations.

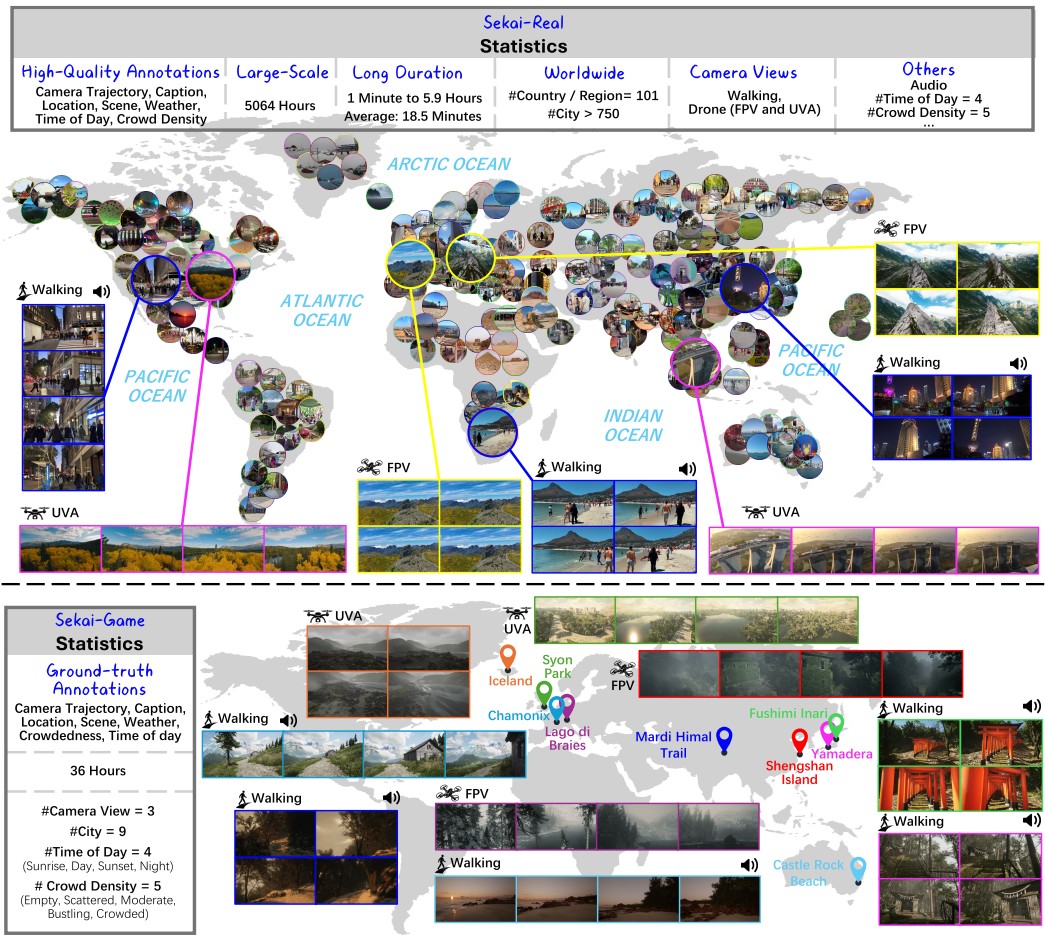

Figure 2: An overview of the Sekai dataset. Sekai-Real is collected from YouTube with high-quality annotations, while Sekai-Game is collected from a game with ground-truth annotations.

- We develop a curation pipeline that efficiently collects, filters, and annotates videos from the web and from video games.
- We validate the quality and effectiveness of the dataset through comprehensive analyses, annotation verification, and experiments on various video generation tasks.

## 2 Related Work

### 2.1 World Generation Model

Recent years have seen a growing interest in video generation [2, 6, 5, 13, 14, 15, 16], 3D scene generation [17, 18, 19, 20, 21, 22], and 4D generation [23, 24, 25, 26, 27], with significant advancements opening up new possibilities in the development of world generation models [28, 29, 30, 31, 32]. In the realm of video generation, text-to-video generation [13, 14] has played a pivotal role, achieving high-fidelity results, while image-to-video generation [15, 16, 33] has also seen notable advancements. Sora [28] further underscores the significance of video generation in the context of world generation models. Among 3D scene generation methods, techniques [19, 20, 21, 17] utilize depth estimation models [34, 35, 36] to extend 2D scenes into 3D representations. 4D scene generation [23, 24, 25] further introduces dynamics, focusing on the evolution of objects or scenes over time [26] and dynamic interactions [27]. This paper primarily focuses on interactive video generation for world exploration, aiming to construct a dynamic and realistic world using image, text, or video for unrestricted exploration.

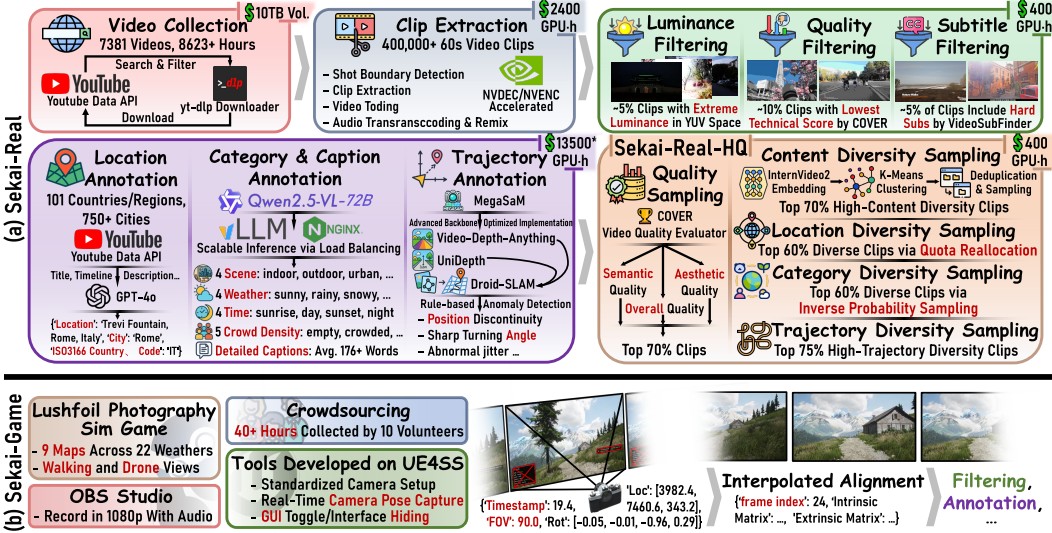

Figure 3: The dataset curation pipeline. *indicates that the statistics were derived from a subset of trajectory annotations.

## 2.2 Video Generation Dataset

The continuous development of annotated datasets has played a pivotal role in shaping the landscape of artificial intelligence-generated content, offering both insights and challenges for accurate model assessment. Existing video generation datasets can be categorized as specific-scenario and open-scenario. Typical specific-scenario datasets including UCF-101 [37], Taichi-HD [38], Sky-Timelapse [39], FaceForensics++ [40], ChronoMagic [41] and Celebv-HQ [42]. These datasets have limited amount of data (with a total duration of less than 800 hours), limited individual video duration (with an average length of less than 20 seconds), and generally lack annotation information (only a few datasets, such as ChromoMagic, provide caption annotations). Open-scenario datasets [43, 44, 45, 46] have somewhat alleviated issues with data scale and annotation information. For example, OpenSoraPlan-V1.0 [45] includes videos with a total duration of 274 hours, each accompanied by detailed captions. Similarly, the recently introduced OpenVid-1M dataset [10] comprises videos totaling 2100 hours, with long captions provided for each video. However, the average duration of individual videos still does not exceed 25 seconds, and they only provide caption annotations. MiraData [12] consists of longer videos with an average length of 72.1 seconds. It is still not long enough for the world exploration, and exploration annotations (*e.g.*, camera poses or keyboard and mouse inputs) and world annotations (*e.g.*, location, time and weather) are missing. By contrast, the proposed Sekai dataset focuses on egocentric world exploration, which covers walking and drone view videos across diverse locations and scenes with long video duration (1 to 39 minutes, average is 2 minutes) and rich exploratory and world annotation.

## 3 Dataset Curation

The overall process of curating the Sekai dataset includes four major parts: video collection, pre-processing, annotation, and sampling, seeing Figure 3 for an illustration.

### 3.1 Video Collection

In the collection stage, we collect over 8623 hours of YouTube videos and over 40 hours of game videos from Lushfoil Photography Sim.

**YouTube**. We manually collect high-quality video URLs from popular YouTubers and extend them by searching additional videos using related keywords (*e.g.*, walk, drone, HDR, and 4K). In total, we collect 10471 hours of walking videos (with stereo audio) and 628 hours of drone (FPV or UAV) videos. All videos were released over the past three years, with a 30-minute to 12-hour duration.

They are at least 1080P with 30 to 60 FPS. We download the 1080P version with the highest Mbps for further video processing and annotation. Due to network issues and some videos are broken, there are 8409 hours of walking videos and 214 hours of drone videos after downloading.

**Video Game**. Beyond real-world data, we collect additional data from the video game since its ground-truth annotations are accessible (*e.g.*, location, weather, and camera trajectory). Lushfoil Photography Sim is a video game that allows walking or using a first-person drone to explore real-world landscapes. It is built by Unreal Engine 5 and showcases the game's locations in stunning visual fidelity, making it an excellent source for collecting realistic synthetic data. We use OBS Studio to record 40 hours videos at 1080P 30FPS (8 to 12 Mbps) with diverse locations and weather. Scaling the amount of data is low-cost.

## 3.2   Video Pre-processing

For YouTube videos, we trim two minutes from the start and end of each original video to remove the opening and ending. Then we do the following steps and obtain 6620 hours (Sekai-Real) and 60 hours (Sekai-Game) of video clips for YouTube and the game, respectively.

**Shot Boundary Detection**. YouTube videos are often cut and stitched, and video games commonly feature teleportation points—both of which contribute to discontinuous shot segments in one video. Thus, following Cosmos [29], we employ TransNetV2 [47] with a threshold of 0.4 for shot boundary detection. However, the original implementation runs slowly. We refactored the codebase for GPU acceleration, which is five times faster than the original version. In particular, we use the PyNVideoCodec library for video decoding and employ the CVCUDA library to offload frame operations such as color space conversion and histogram computation to the GPU. We trim five seconds from the start and end of each shot. After shot detection, the duration of video clips is from 1 to 5.88 hours.

**Clip Extraction and Transcoding**. Considering practical processing, we split each shot into multiple one-minute clips (shorter than one minute will be discarded). In model training, we can stitch contiguous clips according to the computation resources. We re-encode each video clip using the PyNVideoCodec library to standardize the diverse codec configurations in the raw videos, targeting 720p at 30fps in H.265 MP4 format with a bitrate of 4 Mbps. Evaluation of the transcoded video clips across diverse scenes yields PSNR values above 35, indicating no perceptible visual degradation. We think the world exploration should contain realistic sound. Thus, we keep the audio of walking videos. We trim the audio tracks based on the timestamps of the video clips, re-encode them into AAC format using FFmpeg at 48kHz, and mux each audio clip with its corresponding video clip.

**Luminance Filtering**. Overly dark or bright videos are not suitable for model training. We apply a simple filter based on the luma channel in YUV color space, and remove video clips with more than 15 consecutive frames of extremely high or low average brightness. Especially, this step is necessary for video game data, as the engine often employs simplified lighting and camera systems. In this step, we filter out 300 hours of videos.

**Quality Filtering**. We use COVER [48], a comprehensive video quality evaluator to filter low-quality video clips according to the technical quality metric. Technical quality evaluates issues such as image clarity, transmission distortion, and transcoding artifacts. The lowest-scoring 10% of video clips are removed after filtering.

**Subtitle Filtering**. Some videos contain hardcoded subtitles, which are artificial texts embedded in the video frames. These subtitles compromise the video's fidelity to the real world and may introduce misleading patterns during model training. To mitigate this, we apply VideoSubFinder to detect hardcoded subtitles on the bottom one-third of the video frames. A clip is flagged if it contains any subtitle that remains visible for more than 0.75 seconds, in order to reduce false positives. All flagged clips are removed, resulting in the exclusion of approximately 5% of the video clips.

**Camera Trajectory Filtering**. For Sekai-Real, we employ a state-of-the-art structure from motion (SfM) model to extract camera trajectories. However, some trajectories exhibit implausible or counter-intuitive motions, so we heuristically filter out abnormal cases using the following rules. Specifically, we exclude video clips if they satisfy either of the following: (1) Multiple abrupt trajectory reversals (i.e., directional changes exceeding 150 degrees) within a 10-second window. (2) A camera viewpoint shift greater than 60 degrees between two consecutive frames. (3) A camera position displacement

greater than 5 times the average displacement of the 30 consecutive frames containing these two frames. This filtering phase is performed on partially selected data annotated with trajectories.

## 3.3 Video Annotation of Sekai-Real

We annotate the video data from multiple perspectives, including geographic locations, content category and caption, and per-frame camera trajectories.

**Location**. Utilizing the Google YouTube Data API, we fetch the title and description of each video. Since most videos contain multiple chapters filmed at different locations with timeline-based descriptions, we employ GPT-4o [49] to extract a formatted location for each chapter with the ISO 3166 country/region code attached for subsequent processing. We use the interval tree to efficiently match each video clip to its corresponding chapter based on the timestamp, thereby retrieving the location information. Video clips that cannot be uniquely matched to a chapter are discarded, which accounts for approximately 8% of the total clips.

**Category and Caption**. We adopt a two-stage strategy to annotate each video with category and caption. In the first stage, the video is classified along four orthogonal dimensions: scene type, weather, time of day, and crowd density, each with mutually exclusive labels. The model selects the most suitable label for each and abstains when uncertain. In the second stage, we carefully design prompts that incorporate the predicted category labels, location information, and video frames to generate detailed, time-ordered descriptions of actions and scenes for each video clip. Practically, we extract one frame every two seconds from each video clip and use 72B version of Qwen2.5-VL [50] to annotate them. We deploy vLLM [51] inference services with Nginx [52] for load balance. The final caption length averages over 176 words per video clip.

**Camera Trajectories**. We experiment with various camera trajectory annotation methods of different types, including the visual odometry method DPVO [53], the deep visual SLAM framework MegaSaM [34], and a carefully designed 3D transformer VGGT [54] that outputs 3D quantities. Through empirical experiments and comparisons, we choose MegaSaM as the baseline annotation method and made adjustments to optimize annotation accuracy and efficiency. Additionally, we replace the monocular depth estimation model Depth Anything [55] used in MegaSaM with Video Depth Anything [35], which performs better in terms of temporal consistency. We also optimize the official implementation of MegaSaM to support cross-machine, multi-GPU parallel inference, significantly improving annotation efficiency.

## 3.4 Video Annotation of Sekai-Game

We developed a concise yet comprehensive toolchain based on the open-source tools RE-UE4SS and OBS Studio to capture ground-truth annotations from video games. RE-UE4SS is a powerful script system for Unreal Engine, enabling access and modification of the UE object system with minimal overhead at runtime. Based on its Lua Scripting API, we develop practical tools for video collection and annotation, including the standardization of camera system configuration, real-time camera pose capture, GUI hiding, ensuring the collection of clean data with aligned annotations.

The location and category are obtained from the description of the game map, and the prompt used for captioning is tightly modified to better suit the video game context. For camera trajectories, the captured camera poses are further calibrated to compensate for delays and interpolated to synchronize with the video frames.

## 3.5 Video Sampling

Given the prohibitive cost of training on the full Sekai-Real, we propose a strategy to sample the top-tier clips with the highest quality and diversity. The number is related to the computational budget for further video generation model training. In this paper, we sample 400 hours of the videos as Sekai-Real-HQ.

### 3.5.1 Quality Sampling

We sample the highest-quality clips according to two aspects: aesthetic quality and semantic quality. Aesthetic quality reflects the visual harmony among different elements in the video. Semantic quality

assesses the semantic completeness and consistency of the content. We use COVER [48] to obtain two quality scores and sum them for each video clip. We sample a $\alpha_{quality} = 0.7$ proportion of video clips with the highest scores.

### 3.5.2 Diversity Sampling

We balance the videos using the following modules one by one. And for Sekai-Real-HQ, the sampling ratio $\alpha_{content}, \alpha_{loc}, \alpha_{cate}, \alpha_{camera}$ are equal to 70%, 60%, 60%, and 75%, respectivelty.

**Content Diversity**. Given the vast volume of video clips, the presence of similar video clips is inevitable. We use InternVideo2 [56] to extract embeddings for each video clip, and apply mini batch K-Means [57] to cluster the embeddings of each countryregion. Subsequently, in each cluster, we use the scores in quality sampling to rank the samples. Then we iteratively sample a video clip and remove its most similar one until $1 - \alpha_{content}$ proportion of video clips have been removed.

**Location Diversity**. We denote the number of cities as $N_c$. For each city, we count the number of video clips as $N$. Given a sampling ratio $\alpha_{loc}$, we sort the cities in ascending order based on their $N$. For each city in this order, we sample approximately $N \cdot \alpha_{loc}/N$ videos from each city. If it is larger than the corresponding $N$, we sample all video clips for this city and redistribute the shortfall proportionally across the remaining cities by updating $\alpha_{loc}$.

**Category Diversity**. To ensure broad coverage across semantic categories, we perform inverse-probability weighted sampling based on four independent categories: weather, scene, time of day, and crowd density. For each category, we compute the frequency of each label and assign sampling probabilities inversely proportional to their frequencies. Assuming independence among categories, the sampling probability for a video is initialized as the product of its label probabilities across the four categories. These probabilities are then normalized to sum to 1. We perform non-replacement sampling according to these probabilities until $\alpha_{cate}$ proportion of video clips have been sampled.

**Camera Trajectory Diversity**. We perform trajectory-aware sampling by the following steps. First, for the remaining videos, we calculate a direction vector (from the start to end of the trajectory) and the overall jitter, defined as the Euclidean norm of positional variance computed every 30 frames. Next, direction vectors are discretized into bins mapped onto a sphere, and jitter values are also discretized into bins. Then, a joint grouping is formed based on the direction and jitter bins. Finally, we do average sampling in each joint group according to the sampling ratio $\alpha_{camera}$.

## 4 Dataset Statistics

Figure 4 summarizes the statistics of Sekai-Real, which covers 101 countries and regions with a clear long-tail distribution in video duration. The top eight countries (e.g., Japan, the United States, and the United Kingdom) account for about 60% of the total duration. The dataset is categorized by four weather types, four scene types, four time-of-day categories, and five crowd-density levels from various perspectives. Specifically, most videos are outdoor scenes, primarily under sunny or cloudy conditions, while rain and snow further enrich diversity. Daytime footage dominates, followed by nighttime scenes, providing a range of lighting conditions for model learning. Crowd density is evenly distributed, from sparse rural areas to densely populated city streets, supporting tasks such as curriculum learning and evaluation under varying crowd levels. For the Sekai-Game collection, data balance was considered during gameplay.

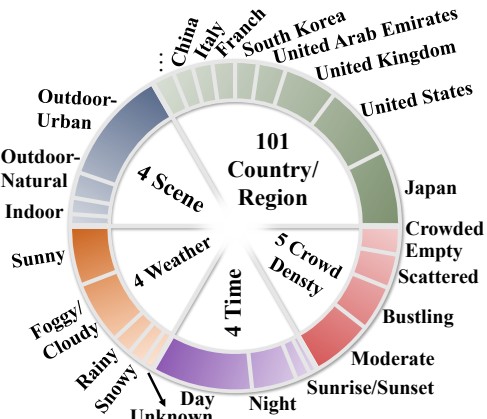

Figure 4: Statistical information on five dimensions of the Sekai-Real dataset.

The statistics of Sekai-Real and Sekai-Real-HQ across multiple dimensions are shown in Figure 5. Sekai-Real-HQ, a top-tier subset of Sekai-Real, features a more balanced data distribution. Seeing Figure 5 (a), Sekai-Real demonstrates strong overall video quality scores, with more than half of

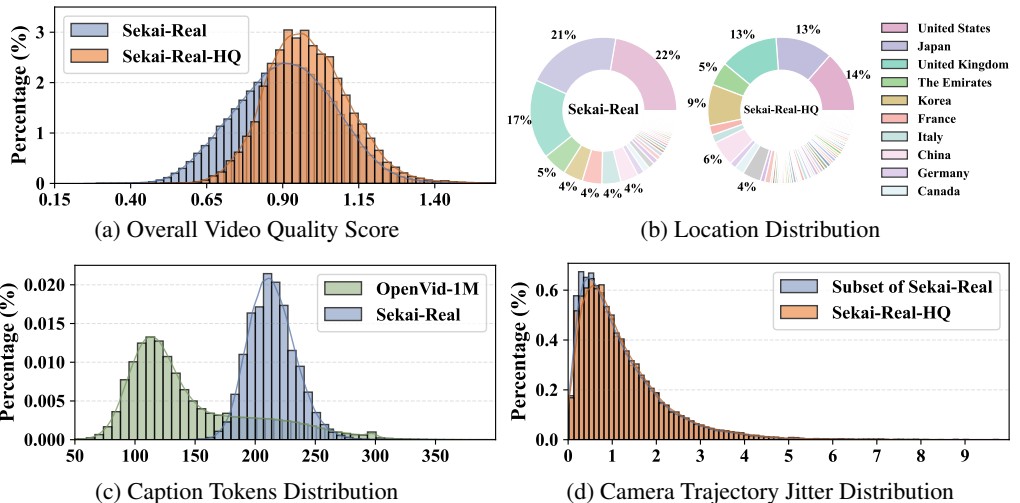

(a) Overall Video Quality Score

(b) Location Distribution

(c) Caption Tokens Distribution

(d) Camera Trajectory Jitter Distribution

Figure 5: Statistics of the proposed Sekai-Real and Sekai-Real-HQ dataset.

the videos scoring above 0.9, while Sekai-Real-HQ exhibits a higher mean video quality score and a lower variance to address the long-tail distribution issue. Figure 5 (b) shows the distribution of Sekai-Real's locations. Both Sekai-Real and Sekai-Real-HQ cover a wide range of countries globally. Sekai-Real-HQ demonstrates a more balanced distribution, which is more effective in mitigating potential bias during model training. In terms of captions, Figure 5(c) shows that Sekai-Real exhibits a higher average token count compared to OpenVid-1M [10], providing richer textual supervision. Figure 5 (d) shows the distribution of camera trajectory jitter before and after applying camera trajectory sampling. We can observe that the distribution for Sekai-Real-HQ is smoother than that of partially selected Sekai-Real data. This indicates that Sekai-Real-HQ achieves better diversity and a more uniform distribution.

## 5 Experiments

In this section, we first validate the quality of the annotation in Sekai. Then we use the top-tier subset Sekai-Real-HQ to fine-tune a video generation foundation model for text-to-video generation and image-to-video generation to validate the effectiveness of the data. Additionally, we explore interactive video generation with the camera trajectories annotated in Sekai.

### 5.1 Evaluation of Annotation Quality

We evaluate the quality of the annotated location, camera trajectory, and category.

**Location Quality**. For videos in Sekai-Real, formatted locations were standardized using GPT-4o [49] based on original YouTube titles and descriptions. To evaluate their quality, we randomly sampled 500 videos and asked co-authors to verify each location using online maps, checking for three possible issues: (1) Omission – missing or incorrectly merged segments; (2) Temporal mismatch – misalignment between the timestamp and location; (3) Location hallucination – inferred rather than explicitly stated locations. Since the title and description provide rich information for GPT-4o, the evaluation revealed that such issues were rare (<5%), indicating that the overall location quality produced by GPT-4o was remarkably high.

**Camera Trajectory Quality**. We use two representative methods of two categories in the field of Structure from Motion, cascaded and end-to-end, namely, MegaSAM [34] and VGGT [54], to annotate 100 walking video clips and 100 drone-view video clips to evaluate their accuracy of camera trajectory prediction. Our validation experiments show that: (1) MegaSAM produces smoother camera trajectories than VGGT, as it has a global optimization module. (2) VGGT offers the advantage of faster inference speed but lower annotation quality. Since data quality is our priority and smoother camera trajectories are more beneficial for model training, we opted for MegaSAM to annotate camera trajectories.

Table 1: Evaluation across training steps for text-to-video generation and image-to-video generation. Higher is better for all metrics.

| Task | Step | I2V Subject | I2V Background | Subject Consistency | Background Consistency | Motion Smoothness | Dynamic Degree | Aesthetic Quality | Imaging Quality | Overall Score |
|------|------|-------------|----------------|---------------------|------------------------|-------------------|----------------|-------------------|-----------------|---------------|
| T2V | 5000 | – | – | 91.93% | 92.41% | **98.40%** | 90.67% | 48.38% | 57.04% | 4.26 |
| | 10000 | – | – | 91.33% | 91.78% | 97.72% | **94.67%** | 48.42% | 60.68% | 4.28 |
| | 15000 | – | – | 93.14% | 93.02% | 97.66% | 89.33% | 49.36% | 60.46% | 4.30 |
| | 20000 | – | – | **96.02%** | **93.58%** | 98.29% | 78.66% | **52.09%** | **60.90%** | **4.34** |
| I2V | 5000 | 93.74% | 93.85% | 87.18% | 90.26% | 97.04% | **100.00%** | 48.76% | 64.84% | 6.10 |
| | 10000 | 96.11% | 95.83% | 90.39% | 90.79% | 97.63% | 98.67% | 49.94% | **68.54%** | 6.26 |
| | 17500 | **97.27%** | **96.65%** | **92.68%** | **91.86%** | **98.70%** | 85.00% | **50.46%** | 66.54% | **6.29** |

**Category Quality**. We randomly sample 500 examples and ask co-authors to label them for evaluating category annotation quality. The sampling strategy follows the category diversity sampling to ensure label variety. The results show that the overall agreement between Qwen2.5-VL and human annotations exceeds 90%. For weather, most discrepancies occur between cloudy/foggy and rainy labels. The difficulty lies in that rain is often imperceptible in a single frame and becomes evident only across consecutive frames, indicating that large vision-language models still struggle with temporal visual reasoning. We plan to leverage dedicated weather prediction datasets to fine-tune video understanding models [56] for more accurate weather annotations. Notably, we exclude indoor scenes from the analysis, as indoor lighting often affects the accuracy of both weather and time-of-day predictions for models and human annotators.

## 5.2 Video Generation

We fine-tune the SkyReels-V2 [58] video generation foundation model on the Sekai-Real-HQ dataset for text-to-video generation and image-to-video generation to validate the effectiveness of the dataset.

### 5.2.1 Settings

**Implementation Details**. We keep the same model architecture and training configurations as those of SkyReels-V2-T2V-14B-540P and SkyReels-V2-I2V-14B-540P for text-to-video and image-to-video generation, respectively. All models are trained with video resolutions of 544×960×49, an FPS of 16, a batch size of 1, and a learning rate of 1e-5. Training was conducted on 8 NVIDIA H100 GPUs for a total of 20,000 iterations. The Adam optimizer is used across all training stages. During inference, we adopt the same resolution and frame rate, with an inference step count of 50.

**Evaluation Dataset**. For a fair evaluation, Sekai-Real-HQ is randomly divided into ten folds, with the last two folds used as the candidate test set and excluded from the training set. Subsequently, independent annotators are invited to manually select 50 clips from the candidate test set, with an emphasis on maintaining diversity during the selection process.

**Evaluation Metrics**. We adopt the VBench [59] evaluation metrics to comprehensively assess the model's performance at different training steps. Specifically, *Subject Consistency* measures whether the subject's appearance remains consistent. *Background Consistency* evaluates the temporal stability of background scenes. *Motion Smoothness* assesses whether motion is smooth and physically plausible. *Dynamic Degree* quantifies the extent of motion to avoid static videos. *Aesthetic Quality* reflects the perceived artistic and visual appeal. Finally, *Imaging Quality* evaluates distortions such as over-exposure, noise, and blur. For the image-to-video generation task, we further adopt the VBench++ [60] metrics *I2V Subject* and *I2V Background* to better evaluate the alignment between the prompt image and the generated video.

### 5.2.2 Quantitative Results

Table 1 presents the results of fine-tuning video generation foundation models in different tasks on the Sekai-Real-HQ. We observed that (1) on both text-to-image generation and image-to-video generation tasks, the overall generation quality improves with training, with consistent gains across most metrics and a steady rise in the *Overall Score*. (2) For image-to-video generation, the *I2V Subject* and *I2V Background* metrics improve markedly, imaging quality increases steadily through training, and the *Overall Score* keeps rising. (3) *Dynamic Degree* decreases after early peaks. This is

Table 2: Evaluation on interactive video generation trained on the drone-view portion of Sekai-Real. Lower is better for *TransErr* and *RotErr*; higher is better for the others.

| Method | TransErr ($\downarrow$) | RotErr ($\downarrow$) | Subject Consistency | Background Consistency | Motion Smoothness | Dynamic Degree | Aesthetic Quality | Imaging Quality |
|--------|---------|--------|---------------------|------------------------|-------------------|----------------|-------------------|-----------------|
| baseline | 28.32 | 27.2 | 97.61% | 94.85% | **99.22%** | 10.67% | 58.25% | 74.73% |
| fine-tuned | **17.19** | **19.89** | **97.34%** | **95.56%** | 99.11% | **10.92%** | **59.18%** | **75.83%** |

a rebalancing where the model shifts from initially exaggerating motion to focusing more on sharper details and cleaner frames, so motion becomes more moderate while overall quality keeps improving.

## 5.3 Interactive Video Generation

We focus on interactive video generation guided by camera trajectories, where the model takes a camera trajectory, an initial image, and a text prompt as inputs to generate a video that follows the defined camera motion. We fine-tune the Wan2.1-Fun-V1.1-1.3B-Control-Camera [5] model with camera trajectories annotated in Sekai-Real and Sekai-Game.

### 5.3.1 Settings

The baseline model uses a rule-based approach to convert discrete camera control action inputs into camera poses, from which it computes Plücker embeddings [61] for each frame and injects them into the model. In contrast, we directly use the per-frame camera poses annotated in Sekai as inputs for fine-tuning. We fine-tune the baseline model for two epochs on the drone-view portion of Sekai-Real and the entire Sekai-Game, respectively. For evaluation, we sample 50 test videos using the same procedure described in the previous section. In addition to the VBench metrics, we adopt two metrics from CameraCtrl [7]: *TransErr* and *RotErr*, which quantitatively evaluate interaction following by measuring the translation and rotation discrepancies between the input trajectory and the trajectory extracted from the generated videos using Mega-SAM.

### 5.3.2 Quantitative Results

Table 2 shows the results of the baseline and the fine-tuned models trained on the drone-view portion of Sekai-Real. The fine-tuned model shows a clear improvement in camera control accuracy, achieving $\triangle 11.13$ reduction in *TransErr* and $\triangle 7.31$ reduction in *RotErr*. Meanwhile, other metrics also show consistent improvements. These results indicate that fine-tuning on Sekai not only improves interaction following but also enhances overall video generation quality in a balanced and comprehensive manner.

Table 3: Evaluation on interactive video generation trained on Sekai-Game. Lower is better.

| Method | TransErr ($\downarrow$) | RotErr ($\downarrow$) |
|--------|---------|--------|
| baseline | 7.64 | 8.36 |
| fine-tuned | **4.22** | **6.22** |

We also fine-tune the baseline model on Sekai-Game, as illustrated in Table 3. The fine-tuned model demonstrates consistent improvements in camera control, with the error rates reduced by more than 30% on average.

## 6 Conclusion

In this paper, we have introduced a new video dataset, Sekai, for video generation–based world exploration. It consists of over 5,000 hours of walking or drone view (FPV and UVA) videos collected from 101 countries and more than 750 cities. We have developed an efficient and effective pipeline to process, filter and annotate the videos. For each video, we annotate location, scene type, weather, crowd density, captions, and camera trajectories. In addition, we present a video sampling module that selects top-tier videos according to the model training budget. Our comprehensive analyses and experiments validate the dataset's scale, diversity, annotation quality, and effectiveness in supporting world exploration video generation model training. We believe that Sekai will benefit the field of video world generation and inspire valuable future applications.

**Acknowledgments** This work was supported by Shanghai Artificial Intelligence Laboratory, Natural Science Foundation of China (NSFC) under No. 62172041 and No. 62176021, Shenzhen Science and Technology Program under Grant No. JCYJ20241202130548062, and Natural Science Foundation of Shenzhen under Grant No. JCYJ20230807142703006.

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
