# OpenReview forum: "Sekai: A Video Dataset towards World Exploration"
_NeurIPS.cc/2025/Datasets_and_Benchmarks_Track — NeurIPS 2025 Datasets and Benchmarks Track poster_

### Official Review · Reviewer_KB8d · 2025-06-18

**Rating:** 5
**Confidence:** 4

**Summary:**

The paper presents Sekai, a “world-exploration” video dataset. Its main components are:
- more than 6,000 hours of first-person walking and drone videos scraped from YouTube (Sekai-Real) plus 36 hours of synthetic footage captured in the UE5 game “Lushfoil Photography Sim” (Sekai-Game).
- Annotation dimensions: location, scene type, weather, time of day, crowd density, text subtitles, and camera trajectories; a portion of the videos also retain their original audio.
- GPU-accelerated shot detection, transcoding, and brightness/quality filtering; multimodal labeling assisted by LLMs and VLMs; a MegaSaM-enhanced camera-trajectory estimator; and a quality- and diversity-aware sampling strategy that produces a 120-hour high-quality subset (Sekai-Real-HQ).
- small-scale manual audits to check annotation accuracy, and fine-tuning SkyReels-V2 on a 13-hour subset for text-to-video and image-to-video experiments, evaluated with VBench.

**Dataset Code Accessibility:**

Yes

**Ethical Considerations:**

No, there are no or only very minor ethics concerns

**Final Justification:**

My primary concern regarding the completeness of the downstream task evaluation has been fully addressed. I recommend that these additional experiments and clarifications be incorporated into the main text. The paper's contribution to the community is unquestionable.

**Limitations Weaknesses:**

1. The downstream validation is limited, as the authors only fine-tune SkyReels-V2 for 13 hours and report six VBench metrics, without any head-to-head comparison against public datasets such as OpenVid-1M or MiraData, leaving Sekai’s unique benefit unproven.
2. Tasks that align more directly with the dataset’s intended use, such as navigation, camera-controlled generation, or egocentric vision-and-language captioning, are missing from the evaluation.
3. Only 300 hours of footage include camera trajectories, and the paper does not provide quantitative metrics such as ATE or RPE for the 200-clip sample, relying instead on qualitative claims of smoother motion.
4. The geographic distribution is long-tailed, with the United States still over-represented; although the HQ subset is balanced by city, the authors do not quantify how this bias might affect world-model training.
5. Label consistency is another concern, as weather-related tags rely on single-frame judgments and show an error rate above fifteen percent in manual checks, with no automatic correction method proposed.
6.The discussion of copyright and compliance is brief and does not fully address YouTube redistribution, creator permissions, or privacy filtering for faces and crowds.

**Strengths Contributions:**

1. The work fits the track well because it is the first dataset that explicitly targets “world exploration” with long, dynamic clips that include trajectory control, a feature needed by interactive video-generation and navigation models.
2. Its scale and diversity are impressive, with roughly 6,000 hours covering 65 countries and more than 1,000 cities, including outdoor and drone points of view, varied weather and day-night conditions, and audio.
3. The dataset offers rich annotations that combine static semantics such as geography and weather with dynamic information such as subtitles and trajectories, thereby enabling multi-task learning and conditional generation.
4. The authors contribute an end-to-end data pipeline that features GPU optimizations and a reusable sampling algorithm, which will likely benefit the community.
5. They apply quality control through manual audits and a dual aesthetic-and-semantic filter to curate the high-quality subset.

---

> ### Author Rebuttal · Authors · 2025-07-31
>
> We are grateful for your time in reviewing our paper and greatly appreciate your valuable suggestions.
>
> > ### **Weakness** **1: Limited downstream validation without comparison to existing datasets.**
>
> **Response**: Thanks for pointing this out. We have conducted more experiments for downstream validation, and the experimental results are shown in the table below. Specifically,  we provide a more comprehensive and unbiased evaluation about text-to-video generation (T2V) and image-to-video (I2V) generation of models trained on over 200 hours of sekai-real-walking-hq data. The evaluation set for either T2V or I2V consists of 50 samples drawn from diverse scenarios. For I2V generation, we additionally incorporate the VBench++ metric to better evaluate alignment with the prompt image. Specifically, Subject consistency measures how consistently the subject in the video matches the input image; Background consistency evaluates the background consistency between the video and the image. We can observe that: (1) as training steps increase, both models show steady performance improvements across nearly all metrics. We will continue training and share updated results during the discussion. (2) The I2V model shows significant gains on two related metrics, highlighting how the diverse data in the Sekai dataset enhances generation quality.
>
> | Type | Training Step | I2V Subject ↑ | I2V Background ↑ | Subject Consistency ↑ | Background Consistency ↑ | Motion Smoothness ↑ | Dynamic Degree ↑ | Aesthetic Quality ↑ | Imaging Quality ↑ |
> | ---- | ------------- | ------------- | ---------------- | --------------------- | ------------------------ | ------------------- | ---------------- | ------------------- | ----------------- |
> | T2V  | 5000  | -  | -   | 91.93%   | 92.41%   | 98.40%   | 90.67%  | 48.38%| 57.04%  |
> |      | 10000  | - | -  | 91.33% | 91.78%  | 97.72%   | 94.67% | 48.42%  | 60.68%  |
> |      | 15000  | -  | -   | 93.14%   | 93.02%     | 97.66%| 89.33% | 49.36%    | 60.46%|
> |      | 20000   | -     | -      | 96.02%     | 93.58%    | 98.29%  | 78.66% | 52.09%   | 60.90%  |
> | I2V  | 5000  | 93.74%| 93.85%   | 87.18%| 90.26%   | 97.04%  | 100% | 48.76%  | 64.84%  |
> |      | 10000  | 96.11%  | 95.83% | 90.39% | 90.79%   | 97.63%   | 98.67%    | 49.94%  | 68.54%  |
> |      | 17500         | 97.27%        | 96.65%           | 92.68%                | 91.86%                   | 98.70%              | 85%              | 50.46%              | 66.54%            |
>
> We would also like to emphasize that Sekai is specifically focused on first-person world exploration, making comparisons with general-purpose datasets such as OpenVID and MiraData inherently unfair. In the next response for Weakness 2, we present experiments on additional downstream tasks, showing that baseline models pre-trained on general datasets perform poorly in world exploration scenarios, while fine-tuning on Sekai significantly enhances their capability in this setting.
>
> ---
>
> > ### **Weakness** **2: Missing evaluation on downstream tasks.**
>
> **Response**: Thank you for your practical suggestion. We have explored camera-controlled video generation and interactive video generation, leveraging the camera trajectories annotated in Sekai.
>
> **Camera-Controlled Video Generation**
>
> Experimental results of camera-controlled video generation are shown in the table below. Specifically, given a camera trajectory , an initial image and a text prompt, Camera-Controlled Video Generation aims to generate an n frame video that follows the specified camera motion while maintaining visual and semantic consistency. Leveraging the annotated camera trajectories, we compute the Plücker embeddings for each frame and inject them into the model during fine-tuning.
>
> Our experiments were conducted on Sekai-Real-Drone based on Wan2.1-Fun-V1.1-1.3B-Control-Camera. We fine-tune the baseline for 2 epochs, and evaluated on a 50 samples test set. We adopt two metrics from CameraCtrl [1]: TransErr and RotErr, to quantitatively evaluate camera control. These metrics measure the discrepancies in translation and rotation between the input camera trajectory and the trajectory extracted from the generated video using Mega-SAM. Experimental results show that camera control error rates decreased by over 30% on average, while all other metrics remained stable or improved. This suggests that fine-tuning on Sekai not only enhances camera control but also improves overall video quality.
>
> | Step       | TransErr ↓ | RotErr ↓ | Subject Consistency ↑ | Background consistency ↑ | Motion Smoothness ↑ | Dynamic Degree ↑ | Aesthetic Quality ↑ | Imaging Quality ↑ |
> | ---------- | ---------- | -------- | --------------------- | ------------------------ | ------------------- | ---------------- | ------------------- | ----------------- |
> | baseline   | 28.32      | 27.2     | 97.61%                | 94.85%                   | 99.22%              | 10.67%           | 58.25%              | 74.73%            |
> | fine-tuned | 17.19      | 19.89    | 97.34%                | 95.56%                   | 99.11%              | 10.92%           | 59.18%              | 75.83%            |
>
> **Interactive Video Generation**
>
> Interactive video generation is a user-friendly and controllable extension of image-to-video (i2v) generation. It takes a sequence of mouse and keyboard inputs to generate videos that reflect user interaction. We find that the recent work YUME [2] fine-tuned SkyReels-V2-14B-540P on our dataset for interactive video generation, demonstrating promising results.
>
> YUME effectively leverages Sekai-Real’s camera trajectory annotations. Their experimental results show that the model trained on Sekai follows user input instruction far better than Wan-2.1 (0.657 vs. 0.057), and significantly outperforms MatrixGame (0.657 vs. 0.271), which targets a similar task but is trained on simpler game data. Moreover, the visualizations provided by YUME show that training with our Sekai dataset further enhances the visual quality of the generated videos.
>
> ---
>
> > ### **Weakness** **3: Limited camera trajectory and no quantitative evaluation.**
>
> **Response**: Camera trajectory annotation is a computationally intensive and time-consuming process. In the submitted version, we provided 300 hours of carefully selected, high-quality camera trajectory annotations. However, the annotation process is still ongoing. To date, we have annotated over 600 hours and aim to complete the annotation for the entire Sekai dataset in the final version.
>
> For the annotated camera trajectories, since we cannot access ground-truth camera trajectories of internet videos, precise quantitative evaluation is impossible. Instead, we manually analyze 50 diverse scenarios across 100 videos and find that VGGT exhibits noticeable jitter in over 60% of cases, as well as observable rotational errors in approximately 20% of glass and mirror scenarios, while MegaSam performs much better. Moreover, VGGT consumes significantly more GPU memory (360GB for 60s videos), making it impractical for Sekai dataset.
>
> ---
>
> > ### **Weakness** **4: Unaddressed geographic bias in dataset distribution.**
>
> **Response**: Geographic bias is inherent and largely unavoidable due to factors such as development, history, and culture. We have actively worked to mitigate this bias by conducting additional data collection in underrepresented regions in the updated dataset. Combined with the tailored location diversity sampling strategy proposed in the paper, these allows us to further reduce geographic bias.
>
> ---
>
> > ### **Weakness** **5:  Inconsistent weather labels with no correction mechanism.**
>
> **Response**: Inconsistent annotations are inevitable, but we have made every effort to minimize such risks and have verified that our dataset maintains over 95% usability. In the latest version, we revised the category definitions to better align with visual characteristics. For example, the weather taxonomy was updated to: Sunny, Foggy/Cloudy, Rainy, and Snowy. The dataset was re-annotated and validated by 10 independent volunteers on 500 samples, achieving 95% agreement. We also plan to use dedicated weather-labeled datasets, such as [4], to train specialized weather annotation models to further improve prediction granularity.
>
> ---
>
> > ### **Weakness 6:  Insufficient discussion of copyright, permissions, and privacy.**
>
> **Response**: We strictly follow the copyright and service agreements of data sources such as YouTube for data collection, annotation, and release. Specifically, we will adopt best practices from prior work, such as Panda-70M, by releasing data in the form of URLs along with corresponding download and processing toolkits, in order to facilitate community while remaining fully compliant with legal and regulatory requirements.
>
> For privacy, we are evaluating several strategies to protect personally identifiable information while minimizing the impact on dataset use and model training. Approaches include face blurring and automated face-replacement anonymization [3]. We will finalize the approach in the camera-ready version.
>
> [1] He, Hao, et al. "Cameractrl: Enabling camera control for video diffusion models." ICLR. 2025.
>
> [2] Mao, Xiaofeng, et al. "Yume: An Interactive World Generation Model." *arXiv2507.17744*.
>
> [3] Hukkelås, Håkon, and Frank Lindseth. "Deepprivacy2: Towards realistic full-body anonymization." WACV. 2023.

---

> > ### Comment · Reviewer_KB8d · 2025-08-03
> > **Further Discussion**
> >
> > Thank you for your detailed response and supplementary experiments addressing the previous review comments.
> >
> > **1. Concerns about the Dynamic Degree metric in T2V/I2V evaluation**
> >
> > We observe that in the first table (T2V/I2V evaluation), a key metric **Dynamic Degree** shows a concerning downward trend.
> >
> > - For T2V, the Dynamic Degree significantly drops from 94.67% at 10k training steps to 78.66% at 20k steps.
> > - For I2V, this metric consistently decreases from 100% at 5k steps to 85% at 17500 steps.
> >
> > This seems to contradict your statement in the rebuttal that "(1) as training steps increase, both models show steady performance improvements across nearly all metrics." We would like to understand the potential reasons for this continuous decline in **Dynamic Degree**.
> > Does this suggest that with longer training, the model tends to generate more static videos with less motion in exchange for improvements in other metrics like "Subject Consistency" or "Background Consistency"?
> > We are concerned this might indicate an inherent trade-off of "sacrificing dynamism for consistency" and would appreciate your deeper analysis on this matter.
> >
> > **2. Discussion on benchmark selection**
> >
> > We understand that Sekai is specifically focused on first-person world exploration, making comparisons with general video datasets like OpenVID potentially unfair. However, we believe that using VBench++ as your primary evaluation benchmark may not be optimal either.
> >
> > - VBench++ is designed to evaluate **general text-to-video/image-to-video generation tasks**, focusing on semantic understanding, visual quality, and action execution.
> > - Given that you explicitly position your task as "**first-person world exploration**", we consider this closer to the domain of **world models** or **embodied AI simulators**.
> >
> > Therefore, we strongly recommend adopting the more specialized **WorldScore Benchmark**[1] for evaluation, as done in the Voyager[2].
> > WorldScore is designed to assess key aspects of generative world models, which are highly relevant to the "world exploration" task.
> > Using WorldScore would provide a more precise measurement of the core capabilities of the Sekai dataset and your model in building coherent, interactive worlds, offering stronger evidence for your work.
> >
> > **3. Questions about camera control experiment setup**
> > Regarding the experiments on camera-controlled video generation in the second part, we have two main concerns about the experimental setup:
> > - **Choice of baseline model for fine-tuning**: You mention that experiments were conducted on the `Wan2.1-Fun-V1.1-1.3B-Control-Camera` model. Our concern is that this model may already be optimized for camera control tasks. For a fairer and clearer demonstration of the value of the Sekai dataset in **introducing and enhancing** camera control capabilities, a more reasonable experimental design would be to **train or fine-tune directly on the original Wan2.1 baseline model** without camera control capabilities. This would more directly prove that it's the Sekai dataset (and its trajectory annotations) that brings performance improvements, rather than achieving marginal gains on an already strong model.
> > - Additionally, you mention a test set containing 50 samples. To better understand the generalizability and robustness of your experimental results, we would appreciate more detailed information about this test set, such as:
> >    * What is the **source** of these samples? Are they a held-out portion of the Sekai dataset or from entirely new, unseen data sources?
> >    * How **diverse** are the samples? Do they cover different types of camera trajectories (e.g., panning, rotation, complex paths), different scenes (indoor, outdoor, urban, natural), and lighting conditions?
> >    * What is the **data distribution** relationship between the test set and the Sekai training data?
> >
> > **4. Concerns about the fairness of comparison in the cited YUME paper**
> > While referencing the YUME paper provides valuable context, we suggest the comparison should be interpreted with caution. The primary concern is that the two systems are evaluated on "instruction-following" using different input modalities, which may not represent a direct, like-for-like comparison.
> > - YUME uses text + keyboard inputs. The keyboard provides direct, explicit, and continuous control signals for motion.
> > - Wan-2.1, as a standard T2V model, relies on text-only input. It must infer complex motion from abstract language, which is a significantly harder task.
> > Therefore, the reported performance gap (0.657 vs. 0.057) is more likely a result of YUME's richer input modality (i.e., the keyboard) rather than the inherent superiority of the Sekai dataset. This comparison does not provide convincing evidence of your dataset's value for interactive generation.
> >
> > [1] WorldScore: A Unified Evaluation Benchmark for World Generation. ICCV 2025.
> >
> > [2] Voyager: Long-Range and World-Consistent Video Diffusion for Explorable 3D Scene Generation. Arxiv 2025.

---

> > > ### Author Response · Authors · 2025-08-08
> > >
> > > Thank you for your valuable feedback. The Sekai dataset was carefully designed with thorough processes for large-scale data collection, multi-dimensional filtering, and rich annotations, all aimed at ensuring high-quality and accurately labeled data. We address the concerns regarding the model training experiments conducted on this dataset in detail.
> > >
> > > > ### **Question 1: The Dynamic Degree metric in T2V/I2V evaluation.**
> > >
> > > Thank you for giving us the opportunity to provide further clarification. **We fully agree with your point, as the "drop" in Dynamic Degree is in line with our expectations.** This represents the learning behavior of the model on the highly dynamic Sekai dataset.
> > >
> > > For deeper analysis, we compare the Dynamic Degree of the baseline models (see below table) and ours fine-tuned model on the test set. We can observe that:
> > >
> > > 1. After only 5,000 training steps (approximately 55 hours of video), the Dynamic Degree increased to 90.67% and 100% for the T2V and I2V baselines, respectively.
> > > 2. Even after the decrease observed during further training, the Dynamic Degree of the fine-tuned models still **exceeded the baselines by 77.33% and 70.33%**.
> > >
> > > We attribute this observation to the highly dynamic nature of the Sekai dataset: during the early stage of fine-tuning, the model tends to maximize motion, focusing on this superficial and easily learnable feature; as training progresses, it gradually improves visual quality while reducing dynamicity to a reasonable level.
> > >
> > > | Model   | **Dynamic Degree (T2V) ↑** | Dynamic Degree (I2V) ↑ |
> > > | :---------------------------- | :------------------------- | :--------------------- |
> > > | baseline (fine-tuned 0 steps) | 1.33%   | 14.67%  |
> > > | Ours (fine-tuned 5000 steps)  | 90.67%   | 100%    |
> > > | Ours (fine-tuned 17500 steps) | 78.66%   | 85.00%  |
> > >
> > > Furthermore, we provide the experimental results using the **Overall Quality Score** from VBench, which is a weighted average of the six dimensions (eight for T2V) described earlier. The experimental results are shown in the table below. It can be observed that the model’s overall generation quality consistently improves throughout the training process, providing further evidence of the usefulness and effectiveness of the proposed Sekai dataset.
> > >
> > > | Type | Training Step | Quality Score |
> > > | ---- | ------------- | ------------- |
> > > | T2V  | 0 (baseline)  | 4.2384  |
> > > |   | 5000   | 4.2639 |
> > > |   | 10000    | 4.282  |
> > > |   | 15000    | 4.2985  |
> > > |   | 20000  | 4.3397  |
> > > | I2V  | 0 (baseline)  | 6.1437  |
> > > |   | 5000 | 6.1045  |
> > > |   | 10000  | 6.2662  |
> > > |   | 17500 | 6.2869 |
> > >
> > > > ### **Question 2: Benchmark selection.**
> > >
> > > Thank you for your suggestion. We aruge that there still **remains a** **large** **gap between the evaluation objectives of WorldScore and the core purpose of world exploration in Sekai.** Nevertheless, the WorldScore results indicate that the models fine-tuned on the Sekai dataset **show improvements** over the baseline models.
> > >
> > > 1. **Gap between WorldScore and Sekai.** WorldScore is a newly released evaluation benchmark (April 2025), introduced close to the NeurIPS submission deadline. Although both WorldScore and Sekai are concerned with world models, WorldScore aims to evaluate general-purpose world generation capabilities, including both photorealistic and stylized scenes. In contrast, Sekai emphasizes first-person world exploration, highlighting a gap between their objectives.
> > > 2. **Experimental results on WorldScore.** WorldScore currently does not support the evaluation of camera-controlled video generation models that take camera trajectories (i.e., camera pose matrices) as input. For that, we evaluated the T2V model fine-tuned on Sekai-Real-Walking-HQ on 300 randomly sampled examples from the benchmark. As shown in the table below, the fine-tuned model achieves stronger performance across most evaluation dimensions, with notable improvements in Camera Control and Object Control.
> > >
> > > |            | Camera Control ↑ | Object Control ↑ | Content Alignment ↑ | 3D Consistency ↑ | Photometric Consistency ↑ | Subjective Quality ↑ | Motion Accuracy ↑ | Motion Magnitude ↑ | Motion Smoothness ↑ |
> > > | ---------- | ---------------- | ---------------- | ------------------- | ---------------- | ------------------------- | -------------------- | ----------------- | ------------------ | ------------------- |
> > > | baseline   | 24.97   | 40.14   | 28.52 | 57.8  | 75.04 | 41.43 | 25.38 | 32.94 | 41.8 |
> > > | fine-tuned | 26.59 | 42.56 | 28.53  | 58.47 | 75.12 | 41.7  | 25.72   | 33.04 | 41.92 |
> > >
> > > Please kindly refer to the next response. We apologize for any inconvenience caused.

---

> > > > ### Author Response · Authors · 2025-08-08
> > > >
> > > > > ### **Question 3.1: Fine-tune the original Wan2.1 model on camera-controlled video generation.**
> > > >
> > > > Thank you for your suggestion. We have now trained directly starting from the weights of the original Wan2.1 I2V baseline model, which does not include camera control capabilities, and **obtained results consistent with our previous findings**. Specifically, we adopted the same model architecture and camera control mechanism as Wan2.1-Fun-V1.1-1.3B-Control-Camera. For the additional parameters introduced in the camera control module of Wan2.1-Fun-V1.1-1.3B-Control-Camera that are not present in the Wan2.1 I2V baseline model, we initialized them randomly using Kaiming initialization.
> > > >
> > > > As shown in the table below, the model trained directly from the original Wan2.1 I2V baseline model (fine-tuned-scratch) **demonstrates better performance in camera control** than previous fine-tuned model (fine-tuned). We deduce this is due to the fact that the Wan2.1-Fun-V1.1-1.3B-Control-Camera model was trained on camera trajectories composed of discrete offsets, as stated in its technical report [1]. For example, a right turn was represented as a fixed offset applied to the camera pose, regardless of the actual angle. Such coarse data limited the benefits of fine-tuning on the high-quality Sekai dataset. In contrast, training from scratch on Sekai, which provides fine-grained and accurate annotations, avoids this limitation and results in more precise camera control. This further emphasizes the importance of high-quality data and the significance of the Sekai dataset.
> > > >
> > > > | Method | TransErr ↓ | RotErr ↓ |
> > > > | ------------------ | ---------- | -------- |
> > > > | baseline | 28.32 | 27.2 |
> > > > | fine-tuned  | 17.19 | 19.89 |
> > > > | fine-tuned-scratch | 7.5 | 9.95 |
> > > >
> > > > > ### **Question 3.2: Details about the test set.**
> > > >
> > > > We adopted a rigorous procedure to construct the test sets used in all experiments. Taking Sekai-Real-Drone as an example (similar procedure for Sekai-Real-Walking), we followed the steps below:
> > > >
> > > > 1. First, we randomly split the raw videos used in Sekai-Real-Drone into 80% for training and 20% as candidate test data. The model is trained exclusively on the former, while the latter serves as a pool for test set selection. This ensures that training and test clips do not come from the same raw video.
> > > > 2. Next, we invited independent annotators to manually select 50 clips from the candidate test set, with an emphasis on diversity during the selection process. The selection was conducted without the annotators viewing any model's predictions.
> > > >
> > > > The final test set exhibits diversity across multiple dimensions, as evidenced by the human evaluation results shown in the table below. Notably, 33% of the test samples involve two or more types of camera motion. Regarding scene categories, the drone-view videos in Sekai-Real-Drone do not contain indoor scenes, while in the test set of Sekai-Real-Walking, 24% test samples are from indoor environments.
> > > >
> > > > | Camera Motion  | Forward | Backward | Leftward | Rightward | Static  |
> > > > | ------------------ | ------- | -------- | -------- | --------- | -------------- |
> > > > | Count  | 35 | 8  | 13  | 15 | 2   |
> > > > | Scene and Lighting | Urban   | Natural  | Daytime  | Nighttime | Sunrise/Sunset |
> > > > | Count   | 29  | 21  | 30  | 7  | 13   |
> > > >
> > > > Please kindly refer to the next response. We apologize for any inconvenience caused.

---

> > > > > ### Author Response · Authors · 2025-08-08
> > > > >
> > > > > > ### **Question 4: Details about the YUME model.**
> > > > >
> > > > > We appreciate your feedback, but we respectfully disagree with your assessments.
> > > > >
> > > > > 1. **Both YUME and Wan-2.1 are standard I2V models**, and they are evaluated using the same input modality (i.e. text), not a richer one for YUME in interactive video generation experiments. Specifically, although Sekai provides fine-grained, continuous camera trajectories, YUME quantizes the camera motion and converts it into text as a form of textual condition (see Section 5.4 of their paper for details). Therefore, during evaluation, both YUME and Wan-2.1 take the same input: a first-frame image and a text prompt, where the prompt includes the text description of the camera motion. Both models are required to infer complex motion from abstract language, which is a significantly more challenging task. YUME’s clearly superior performance highlights the value of our dataset for interactive video generation.
> > > > > 2. **MatrixGame model, which relies on direct keyboard input, still performs significantly worse than YUME.** The YUME paper also compares its method to MatrixGame, which uses additional keyboard inputs. The keyboard provides direct, explicit, and continuous control signals for motion. In contrast, YUME must infer motion from abstract language alone, making the task considerably harder. Yet, YUME still achieves a much higher instruction-following score (0.657 vs. 0.271). This comparison, which is unfavorable to YUME in terms of input richness, further emphasizes the intrinsic advantages of the Sekai dataset and provides strong, undeniable evidence of its value in interactive generation.
> > > > > 3. **YUME's applications to other tasks further highlight the broad applicability and value of Sekai**. The YUME paper demonstrates that models trained on the Sekai dataset benefit in various downstream tasks, such as long video generation, world generalization, and world editing.
> > > > >
> > > > > We have contacted the authors of YUME and confirmed the accuracy of the aforementioned statement. We remain open to further dialogue and welcome your suggestions on whether our revisions are satisfactory.
> > > > >
> > > > > [1] Wan, Team, et al. "Wan: Open and advanced large-scale video generative models." arxiv. 2025.

---

> > > > > > ### Comment · Reviewer_KB8d · 2025-08-08
> > > > > >
> > > > > > I thank the authors for their diligent efforts in the rebuttal. My primary concern regarding the completeness of the downstream task evaluation has been fully addressed. I recommend that these additional experiments and clarifications be incorporated into the main text. The paper's contribution to the community is unquestionable, and I will raise my score accordingly.

---

> > > > > > > ### Author Response · Authors · 2025-08-08
> > > > > > >
> > > > > > > Thank you for your positive feedback. We will revise the final version accordingly, taking into account your suggestions and incorporating the additional experiments and clarifications.

---

### Official Review · Reviewer_Acq5 · 2025-06-27

**Rating:** 5
**Confidence:** 4

**Summary:**

The paper introduces Sekai, a high-quality first-person view worldwide video dataset with rich annotations. It comprises two subsets: Sekai-Real, containing over 6,000 hours of high-quality walking and drone videos from YouTube across 65 countries and 1,000+ cities, and Sekai-Game, with 36 hours of synthetic, ground-truth-annotated footage from the Lushfoil Photography Sim game. Each clip is at least one minute long (up to 39 minutes), 1080p@30 FPS, and richly annotated with location, scene type, weather, crowd density, captions (∼200 tokens), audio, and camera trajectories.

The authors detail a GPU-accelerated curation pipeline (shot detection, transcoding, filtering), an LLM-driven annotation framework, and a sampling module to produce Sekai-Real-HQ, a balanced 120 hour subset. They validate annotation accuracy (<5% location errors; >85% category agreement), and demonstrate dataset utility by fine-tuning SkyReels-V2 models for text-to-video and image-to-video generation, showing consistent improvements with more training steps.

**Additional Feedback:**

I suggest the authors extend their evaluation—either by training on a larger portion of the dataset and testing on a more substantial held-out set, or by providing a stronger justification for why fine-tuning on just 13 hours of video and evaluating on only 14 sampled clips is sufficient to demonstrate the dataset’s effectiveness.

**Dataset Code Accessibility:**

Yes

**Dataset Code Comments:**

The authors provide links and code in their supplementary, making it easy to access to the dataset and code.

**Ethical Considerations:**

No, there are no or only very minor ethics concerns

**Final Justification:**

The authors have done lots of evaluation to justify their dataset. After reviewing all the responses from my discussion and discussions of other reviewers, I believe it is a useful dataset for the field of video generation and I think this paper could be accepted.

**Limitations Weaknesses:**

Annotation Subset: As mentioned in the limitation section in the paper, the camera trajectories are only annotated for a 300 h subset of Sekai-Real-HQ due to compute limits, but I am sure the authors are trying to scale it up after the submission.

Limited Model Evaluation: Experiments fine-tune SkyReels-V2 on only 13h of data and only sampled 14 testing videos for evaluation on Vbench.

Under-explored Modalities (Minor): The role of audio annotations is noted but not empirically evaluated, missing an opportunity to showcase multimodal dataset strengths.

**Strengths Contributions:**

Scale and Diversity: With over 6,000 hours spanning 65 countries, 1000+ cities, diverse weathers, time of day, and crowd densities, Sekai greatly exceeds prior datasets in both duration and world coverage.

Methodological Rigor: The curation pipeline demonstrates both technical novelty and engineering effort.

Reproducibility: The authors open-sourced dataset, provide URLs, and detailed resource tables (e.g., GPU hours) to facilitate dataset reconstruction.

---

> ### Author Rebuttal · Authors · 2025-07-31
>
> Thank you for recognizing the significance of my work and for your constructive comments, which will help enhance the quality of my paper.
>
> > ### **Weakness** **1: Limited camera trajectory annotation subset.**
>
> **Response**: Thank you for recognizing our work. Camera trajectory annotation is a computationally intensive and time-consuming process. In the submitted version, we provided 300 hours of carefully selected, high-quality camera trajectory annotations. However, the annotation process is still ongoing. To date, we have annotated over 600 hours and aim to complete the annotation for the entire Sekai dataset in the final version.
>
> ---
>
> > ### **Weakness** **2: Limited model training and evaluation.**
>
> **Response**: We have increased the training data scale and conduct more experiments, and the experimental results are shown in the table below. Specifically, we extend the training to over 200 hours of sekai-real-walking-hq data, and reconstructing the test set with 50 test samples randomly selected from diverse scenarios to ensure a more comprehensive and unbiased assessment. For image-to-video generation, we additionally incorporate the VBench++ metric I2V Subject and I2V Background to better evaluate alignment with the prompt image. We can observe that: (1) as training steps increase, both models show steady performance improvements across nearly all metrics. We will continue training and share updated results during the discussion. (2) The I2V model shows significant gains on two related metrics, highlighting how the diverse data in the Sekai dataset enhances generation quality.
>
> | Type | Training Step | I2V Subject ↑ | I2V Background ↑ | Subject Consistency ↑ | Background Consistency ↑ | Motion Smoothness ↑ | Dynamic Degree ↑ | Aesthetic Quality ↑ | Imaging Quality ↑ |
> | ---- | ------------- | ------------- | ---------------- | --------------------- | ------------------------ | ------------------- | ---------------- | ------------------- | ----------------- |
> | T2V  | 5000          | -             | -                | 91.93%                | 92.41%                   | 98.40%              | 90.67%           | 48.38%              | 57.04%            |
> |      | 10000         | -             | -                | 91.33%                | 91.78%                   | 97.72%              | 94.67%           | 48.42%              | 60.68%            |
> |      | 15000         | -             | -                | 93.14%                | 93.02%                   | 97.66%              | 89.33%           | 49.36%              | 60.46%            |
> |      | 20000         | -             | -                | 96.02%                | 93.58%                   | 98.29%              | 78.66%           | 52.09%              | 60.90%            |
> | I2V  | 5000          | 93.74%        | 93.85%           | 87.18%                | 90.26%                   | 97.04%              | 100%             | 48.76%              | 64.84%            |
> |      | 10000         | 96.11%        | 95.83%           | 90.39%                | 90.79%                   | 97.63%              | 98.67%           | 49.94%              | 68.54%            |
> |      | 17500         | 97.27%        | 96.65%           | 92.68%                | 91.86%                   | 98.70%              | 85%              | 50.46%              | 66.54%            |
>
> ---
>
> > ### **Weakness** **3: Under-explored audio modalities.**
>
> **Response**: Thanks for pointing this out. Given that human perception relies on both vision and sound, generating realistic audio is a key capability of an ideal world model. In video generation, sota models like Veo 3 and the newly released Wan2.2 have begun exploring video generation with audio, underscoring the forward-looking design of our dataset. With rich annotations including audio, location data, and camera trajectories, our dataset offers broad utility and research potential.

---

> > ### Comment · Reviewer_Acq5 · 2025-08-03
> >
> > Thank you for addressing my concerns. Your response have resolved my questions, and I will maintain my original score.

---

### Official Review · Reviewer_UV8k · 2025-06-30

**Ethics Flags:** Data privacy, copyright, and consent
**Rating:** 4
**Confidence:** 4

**Summary:**

This paper introduces Sekai, a large-scale video dataset designed for world exploration research. The dataset comprises two parts: Sekai-Real, with over 6000 hours of high-quality, first-person walking and drone videos collected from 65 countries on YouTube, and Sekai-Game, a smaller set of videos from a photorealistic video game with ground-truth annotations. The dataset contains a rich set of annotations, including camera trajectories, location, scene type, weather, crowd density, and detailed captions.

**Additional Feedback:**

N/A

**Dataset Code Accessibility:**

Yes

**Dataset Code Comments:**

The dataset and the full code for their pipeline were accessible.

**Ethical Comments:**

Since the dataset includes first-person walking videos captured in over 1000 cities, the videos are highly likely to contain personally identifiable information, such as faces. The paper does not mention any process to detect and filter out (or blur) that information from dataset.

**Ethical Considerations:**

Yes, there are ethics concerns that require attention by the authors

**Final Justification:**

The authors' rebuttal has convincingly addressed my initial concerns. The authors conducted experiments for sufficient validation of the proposed dataset. Thus, I have raised my score accordingly.

**Limitations Weaknesses:**

1. Insufficient experimental validation: The quantitative results provided are not sufficient to fully assess the quality and utility of this large-scale dataset. The video generation experiments were conducted on only 13 hours of data, a minuscule fraction of the full 6000 hour dataset. Such a small-scale experiment, while showing a positive trend, is not robust enough to demonstrate the true potential of this massive resource. Crucially, a key advertised contribution, the camera trajectory annotations, was not used to train or evaluate a trajectory-controlled model, representing a significant missed opportunity to showcase the dataset's unique capabilities. Furthermore, there is no evaluation for Sekai-Game in the paper.

2. Annotation inaccuracy and inconsistency:  The dataset appears to contain annotations that are either inaccurate or inconsistent, which could affect its usability. The paper's evaluation of annotation quality was performed by co-authors, not independent annotators, which introduces subjectivity to the reported accuracy figures (e.g., >85% agreement for categories, <5% error for locations). The location annotations generated by GPT-4o can be inconsistent in their format and granularity. For example, there are lots of samples with “unspecified” locations, which is not appropriate for world exploration.

3. Limited scope of annotations: Camera trajectories are highlighted as a core feature for world exploration, intended to address a key limitation in prior work. However, due to high computational costs, these annotations are only available for a 300-hour subset of the data. The fact that this critical exploratory annotation is missing from the vast majority (~95%) of the Sekai-Real dataset is a significant limitation.

**Strengths Contributions:**

1. The proposed dataset, Sekai, contains 6000 hours of footage, including videos captured from various countries, cities, weather, times of day, and crowd densities.

2. The paper provides a detailed and transparent account of the dataset curation pipeline.

3. The paper is well-written, organized, and easy to follow. The figures are highly informative, and the charts effectively illustrate the statistical properties.

---

> ### Author Rebuttal · Authors · 2025-07-31
>
> Thank you for your comments. We have conducted more experiments including: (1) scaling up the training data for training T2V and I2V; (2) training models for camera trajectory-guided video generation in drone view using the Sekai-real dataset; and (3) training similar models using the Sekai-game dataset.
>
> > ### **Weakness** **1.1: Limited scale of validation.**
>
> **Response**: We have scale up the training data to conduct more experiments, and the experimental results are shown in the table below. Specifically, we extend the training to over 200 hours of sekai-real-walking-hq, and reconstructing the test set with 50 test samples randomly selected from diverse scenarios to ensure a more comprehensive and unbiased assessment. For image-to-video generation, we additionally incorporate the VBench++ metric I2V Subject and I2V Background to better evaluate alignment with the prompt image. We can observe that: (1) as training steps increase, both models show steady performance improvements across nearly all metrics. We will continue training and share updated results during the discussion. (2) The I2V model shows significant gains on two related metrics (I2V Subject and I2V Background), highlighting how the diverse data in the Sekai dataset enhances generation quality.
>
> | Type | Training Step | I2V Subject ↑ | I2V Background ↑ | Subject Consistency ↑ | Background Consistency ↑ | Motion Smoothness ↑ | Dynamic Degree ↑ | Aesthetic Quality ↑ | Imaging Quality ↑ |
> | ---- | ------------- | ------------- | ---------------- | --------------------- | ------------------------ | ------------------- | ---------------- | ------------------- | ----------------- |
> | T2V  | 5000          | -     | -                | 91.93%                | 92.41%                   | 98.40%              | 90.67%           | 48.38%              | 57.04%            |
> |      | 10000         | -      | -        | 91.33%                | 91.78%                   | 97.72%              | 94.67%           | 48.42%              | 60.68%            |
> |      | 15000         | -             | -                | 93.14%                | 93.02%                   | 97.66%              | 89.33%           | 49.36%              | 60.46%            |
> |      | 20000         | -             | -                | 96.02%                | 93.58%                   | 98.29%              | 78.66%           | 52.09%              | 60.90%            |
> | I2V  | 5000          | 93.74%        | 93.85%           | 87.18%                | 90.26%                   | 97.04%              | 100%             | 48.76%              | 64.84%            |
> |      | 10000         | 96.11%        | 95.83%           | 90.39%                | 90.79%                   | 97.63%              | 98.67%           | 49.94%              | 68.54%            |
> |      | 17500         | 97.27%        | 96.65%           | 92.68%                | 91.86%                   | 98.70%              | 85%              | 50.46%              | 66.54%            |
>
> ---
>
> > ### **Weakness** **1.2: Camera trajectory annotations not utilized in experiments**
>
> **Response**: Thanks for pointing this out. We have explored camera-controlled video generation and interactive video generation, leveraging the camera trajectories annotated in Sekai.
>
> **Camera-Controlled Video Generation**
>
> Experimental results of camera-controlled video generation are shown in the table below. Specifically, given a camera trajectory  , an initial image and a text prompt, Camera-Controlled Video Generation aims to generate an $n$ frame video that follows the specified camera motion. Leveraging the annotated camera trajectories in Sekai, we compute the Plücker embeddings for each frame and inject them into the model during fine-tuning.
>
> Our experiments were conducted on Sekai-Real-Drone based on Wan2.1-Fun-V1.1-1.3B-Control-Camera. If you are interested in larger-scale results or more details, we welcome your questions during the discussion. We fine-tune the baseline for 2 epochs, and evaluated on a test set containing 50 randomly selected samples from diverse scenarios. We adopt two metrics from CameraCtrl [1]: TransErr and RotErr, to quantitatively evaluate camera control. These metrics measure the discrepancies in translation and rotation between the input camera trajectory and the trajectory extracted from the generated video using Mega-SAM. Experimental results show that camera control error rates decreased by over 30% on average, while all other metrics remained stable or improved. This suggests that fine-tuning on Sekai not only enhances camera control but also improves overall video quality.
>
> | Step       | TransErr ↓ | RotErr ↓ | Subject Consistency ↑ | Background consistency ↑ | Motion Smoothness ↑ | Dynamic Degree ↑ | Aesthetic Quality ↑ | Imaging Quality ↑ |
> | ---------- | ---------- | -------- | --------------------- | ------------------------ | ------------------- | ---------------- | ------------------- | ----------------- |
> | baseline   | 28.32      | 27.2     | 97.61%                | 94.85%                   | 99.22%              | 10.67%           | 58.25%              | 74.73%            |
> | fine-tuned | 17.19      | 19.89    | 97.34%                | 95.56%                   | 99.11%              | 10.92%           | 59.18%              | 75.83%            |
>
> **Interactive Video Generation**
>
> Interactive video generation is a user-friendly and controllable extension of image-to-video (i2v) generation. It takes a sequence of mouse and keyboard inputs to generate videos that reflect user interactions, similar to controlling movement in 3D games. We find that the recent work YUME [2] fine-tuned SkyReels-V2-14B-540P on our dataset for interactive video generation, demonstrating promising results.
>
> YUME effectively leverages Sekai-Real’s camera trajectory annotations. It quantizes continuous trajectories into discrete motion types (e.g., move forward, turn left) by matching each relative pose to a canonical template, extracts three motion descriptors, and converts them into text tokens combined with the user's prompt. Their experimental results show that the model trained on Sekai follows user input instruction far better than Wan-2.1 (0.657 vs. 0.057), and significantly outperforms MatrixGame (0.657 vs. 0.271), which targets a similar task but is trained on simpler game data. Moreover, the visualizations provided by YUME show that training with our Sekai dataset further enhances the visual quality of the generated videos.
>
> ---
>
> > ### **Weakness** **1.3: Missing evaluation of Sekai-Game.**
>
> **Response**: Thank you for your comments. We have conducted experiments about camera-controlled video generation using Sekai-Game, with the results shown in the table below. The results demonstrate that ground-truth trajectory annotations from Sekai-Game significantly improve camera control (△3.42 on TransErr and △2.14 on RotErr) and increase the dynamic degree, producing more dynamic videos while maintaining performance comparable to the baseline on other metrics.
>
> | Step       | TransErr ↓ | RotErr ↓ | Subject Consistency ↑ | Background consistency ↑ | Motion Smoothness ↑ | Dynamic Degree ↑ | Aesthetic Quality ↑ | Imaging Quality ↑ |
> | ---------- | ---------- | -------- | --------------------- | ------------------------ | ------------------- | ---------------- | ------------------- | ----------------- |
> | baseline   | 7.64       | 8.36     | 93.03%                | 93.06%                   | 99.02%              | 86.00%           | 51.52%              | 45.04%            |
> | fine-tuned | 4.22       | 6.22     | 88.87%                | 93.63%                   | 98.54%              | 100%             | 50.11%              | 41.79%            |
>
> ---
>
> > ### **Weakness** **2: Inaccuracy and inconsistency annotation.**
>
> **Response**: Inaccurate and inconsistent annotations are inevitable, but we have made every effort to minimize such risks and have verified that our dataset maintains over 95% usability. In the latest version, we revised the category definitions to better align with visual characteristics. For example, the weather taxonomy was updated to: Sunny, Foggy/Cloudy, Rainy, and Snowy. The dataset was re-annotated and validated by 10 independent volunteers on 500 samples, achieving 95% agreement. We also refined the location metadata by removing samples lacking sub-city detail. Locations are now structured as: [detailed place]–[city]–[country].
>
> ---
>
> > ### **Weakness** **3: Limited scope of camera trajectory annotations.**
>
> **Response**: Camera trajectory annotation is a computationally intensive and time-consuming process. In the submitted version, we provided 300 hours of carefully selected, high-quality camera trajectory annotations. However, the annotation process is still ongoing. To date, we have annotated over 600 hours and aim to complete the annotation for the entire Sekai dataset in the final version.
>
> ---
>
> > ### **Ethical Comments 1: Personally identifiable information, such as faces.**
>
> **Response**: Thank you for the valuable comments. We are evaluating several strategies to protect personally identifiable information while minimizing the impact on dataset use and model training. Approaches include face blurring and automated face-replacement anonymization [3]. We will finalize the approach in the camera-ready version.
>
> [1] He, Hao, et al. "Cameractrl: Enabling camera control for video diffusion models." ICLR. 2025.
>
> [2] Mao, Xiaofeng, et al. "Yume: An Interactive World Generation Model." *arXiv2507.17744*.
>
> [3] Hukkelås, Håkon, and Frank Lindseth. "Deepprivacy2: Towards realistic full-body anonymization." WACV. 2023.

---

> > ### Comment · Reviewer_UV8k · 2025-08-05
> >
> > I thank the authors for their detailed rebuttal. They have effectively addressed all the major weaknesses I pointed out in my initial review, but I have additional question about the rebuttal.
> >
> > Regarding the rebuttal for Weakness 1.3, I see that fine-tuning on Sekai-Game improves camera control and dynamic degree. However, the table shows that other metrics like Subject Consistency and Aesthetic/Imaging Quality worsened. Could you explain this trade-off?

---

> > > ### Author Response · Authors · 2025-08-08
> > >
> > > Thank you for your valuable feedback. We observe this trade-off is widely observed across existing SOTA video generation models and can be effectively mitigated on the Sekai-Game by increasing the training to 7 epochs.
> > >
> > > 1. **SOTA models exhibit similar or even more pronounced trade-offs on Vbench**. We observe from the VBench Leaderboard [1] provided by the VBench team that models achieving SOTA performance on Dynamic Degree typically lag behind the corresponding SOTA models in Subject Consistency and Aesthetic/Imaging Quality by an average of 8.8%. Conversely, models that perform best in Subject Consistency and Aesthetic/Imaging Quality fall short on Dynamic Degree by an average of 53%.
> > > 2. **Quality-related metrics exhibit clear improvment after extending the fine-tuning to 7 epochs.** As shown in the table below, the model achieves improvements in Subject Consistency and Aesthetic/Imaging Quality while maintaining performance on Dynamic Degree, with particularly notable gains in Imaging Quality. This behavior can be attributed to the highly dynamic characteristics of the Sekai dataset. In the early stage of fine-tuning, the model prioritizes maximizing motion, as it is a prominent and easily captured feature. As training continues, the model shifts its focus toward enhancing visual quality while maintaining dynamicity.
> > > 3. Benefiting from the high dynamism and accurate ground-truth annotations of the Sekai-Game dataset, these fine-tuned models achieve 100% on **Dynamic Degree, reaching the upper bound of the VBench metric**. This suggests that a more suitable evaluation metrics, dimension, capabilities, and benchmark is needed to properly assess world exploration capabilities, which we consider an important direction for future work.
> > >
> > > | Method             | Subject Consistency ↑ | Background consistency ↑ | Motion Smoothness ↑ | Dynamic Degree ↑ | Aesthetic Quality ↑ | Imaging Quality ↑ |
> > > | ------------------ | --------------------- | ------------------------ | ------------------- | ---------------- | ------------------- | ----------------- |
> > > | baseline           | 93.03%                | 93.06%                   | 99.02%              | 86.00%           | 51.52%              | 45.04%            |
> > > | fine-tuned(2epoch) | 88.87%                | 93.63%                   | 98.54%              | 100%             | 50.11%              | 41.79%            |
> > > | fine-tuned(7epoch) | 89.47%                | 93.69%                   | 98.55%              | 100%             | 51.12%              | 44.87%            |
> > >
> > > We hope to hear your thoughts on whether our responses adequately resolve your concerns.
> > >
> > > [1] Huang, Ziqi, et al. "Vbench: Comprehensive benchmark suite for video generative models." CVPR. 2024.

---

> > > > ### Comment · Reviewer_UV8k · 2025-08-09
> > > >
> > > > Thank you for the clarification. I hope the authors include these experiments and the analysis of the performance trade-off in the final manuscript. As all of my concerns have now been resolved, I will raise my score.

---

### Official Review · Reviewer_6t18 · 2025-07-06

**Rating:** 4
**Confidence:** 4

**Summary:**

1. This paper introduces a large-scale video dataset designed for “world exploration” tasks.
2. It comprises 6,000 hours of first-person and drone footage collected from 65 countries.
3. Each video is annotated with geographic location (via the YouTube API), scene category (provided by Qwen), descriptive captions, and high-precision camera trajectories estimated using MegaSaM.

**Dataset Code Accessibility:**

Yes

**Dataset Code Comments:**

I find the preprocess code in the supplementary and their huggingface page is clear.

**Ethical Considerations:**

No, there are no or only very minor ethics concerns

**Final Justification:**

While I find the current data annotation methods to be less than satisfactory, particularly in terms of their reliability, I am not fully convinced by the authors' attempt to demonstrate this through their evaluations. Therefore, I can only offer a "weak accept".

**Limitations Weaknesses:**

1. Unclear annotation quality evaluation. Section 5.1.2 describes running VGGT and MegaSaM on the data but does not report quantitative metrics or detailed conclusions about annotation accuracy. It remains unclear how the authors validate or benchmark the quality of each annotation type.
2. In Section 5.2.3, the authors present metrics for a video generation model fine-tuned on the raw Sekai dataset, but they never define what each metric actually measures, nor do they show the model’s baseline performance before fine-tuning for comparison. The authors should therefore clearly explain how each metric is computed and what aspect of generation it reflects, present side-by-side before-and-after results, and analyze why fine-tuning on Sekai produces these specific shifts in performance.

**Strengths Contributions:**

1. Scale and pose annotation. The release of a 6,000-hour, pose-annotated open dataset fills a crucial gap. It will not only enable video-based world exploration models but also accelerate advances in feed-forward 3D reconstruction. For example, the authors show that MegaSaM often outperforms VGGT, suggesting that models trained on this dataset could surpass existing VGGT-based methods.

2. Technically sound annotation pipeline. The dataset’s metadata are generated via a automated workflow—leveraging the YouTube API, Qwen for semantic tagging, and MegaSaM for trajectory estimation—ensuring consistency and reliability.

---

> ### Author Rebuttal · Authors · 2025-07-31
>
> We truly appreciate the time and effort you devoted to reviewing our paper, as well as your recommendation for acceptance and your insightful comments and feedback.
>
> > ### **Weakness** **1: Quantitative evaluation or detailed conclusions about VGGT and MegaSaM.**
>
> **Response**:  Thank you for your comments. We would like to present more evaluation details and analysis here. Since we cannot access ground-truth camera trajectories of internet videos, precise quantitative evaluation is impossible. Instead, we manually analyze 50 diverse scenarios (e.g., weather, crowd density, glass) across 100 videos and find that VGGT exhibits noticeable jitter in over 60% of cases, as well as observable rotational errors in approximately 20% of glass and mirror scenarios, while MegaSam performs much better. Moreover, VGGT consumes significantly more GPU memory (360GB for 60s videos), making it impractical for Sekai dataset.
>
> ---
>
> > ### **Weakness** **2: Missing metric definitions and baseline comparisons for fine-tuning evaluation.**
>
> **Response**: Thank you for your comments. We would like to present the definitions of metrics introduced by VBench and show the experimental comparsions with the baseline model after training on over 200 hours of Sekai-Real-Walking-HQ data.
>
> Metric definitions:
>
> - Subject Consistency measures whether the subject’s appearance remains consistent;
> - Background Consistency evaluates the temporal stability of background scenes;
> - Motion Smoothness assesses whether motion is smooth and physically plausible;
> - Dynamic Degree measures the extent of motion to avoid static videos;
> - Aesthetic Quality reflects the perceived artistic and visual appeal;
> - Imaging Quality evaluates distortions such as over-exposure, noise, and blur.
>
> Comparisions with the baseline model: The table below shows the comparison results on the image-to-video generation task. The model fine-tuned on Sekai outperforms the baseline across all metrics, with substantial improvements in Subject/Background Consistency and Imaging Quality, demonstrating superior visual fidelity.
>
> | Model   | Subject Consistency ↑ | Background Consistency ↑ | Motion Smoothness ↑ | Aesthetic Quality ↑ | Imaging Quality ↑ |
> | ------- | --------------------- | ------------------------ | ------------------- | ------------------- | ----------------- |
> | Wan-2.1 | 0.859                 | 0.899                    | 0.961               | 0.494               | 0.695             |
> | Ours    | 0.932                 | 0.941                    | 0.986               | 0.518               | 0.739             |

---

> > ### Comment · Reviewer_6t18 · 2025-08-05
> >
> > We know that online videos lack pose annotations, making evaluation difficult. However, I believe this is the responsibility of the authors. They need to devise alternative methods to convince everyone that their annotation methods are reliable, or how reliable they are. For example, they could apply their annotation methods to in-the-wild videos containing ground-truth poses; or they could define some done-stream tasks, such as view synthesis checks, to inform everyone about the effectiveness of current annotation methods.

---

> > > ### Author Response · Authors · 2025-08-08
> > >
> > > Thank you for your suggestion. The reliability of our video pose annotation method can be supported in the following aspects:
> > >
> > > 1. The pose annotation method (i.e., MegaSam) we use is currently state-of-the-art. This is not only demonstrated by the empirical results we previously discussed, but also **supported by the quantitative experiments** presented in [1]. In that work, MegaSAM and VGGT are compared on three typical benchmarks for the task of camera pose estimation, including TUM Dynamics [2], Lightspeed [3], and Sintel [4], as shown in the table below. Specifically, TUM Dynamics is a in the wild dataset captured using well-calibrated RGB-D sensors, while Sintel and Lightspeed are synthetic datasets that feature numerous challenging dynamic scenes with large ego-motion and significant object motions. We can observe that MegaSAM consistently outperforms VGGT across various scenes and motion conditions, exhibiting significantly lower errors in both absolute trajectory error and relative pose error, resulting in more reliable camera trajectories in Sekai. Furthermore, the quantitative results on the Video Depth Evaluation task in [1] also demonstrate the superiority and reliability of MegaSAM. The experimental details can be found in Section 4.2.2 of [1].
> > >
> > > | Benchmark     | TUM-dynamic                   | Lightspeed                        | Sintel                            |
> > > | ------------- | ----------------------------- | --------------------------------- | --------------------------------- |
> > > | Error Rate(↓) | ATE / RPE_t / RPE_r           | ATE / RPE_t / RPE_r               | ATE / RPE_t / RPE_r               |
> > > | VGGT          | 0.021 / 0.013 / 0.327         | 0.226 / 0.086 / 1.729             | 0.082 / 0.043 / 1.253             |
> > > | MegaSAM       | **0.013** / **0.011** / 0.340 | **0.105** / **0.040** / **0.996** | **0.023** / **0.008** / **0.060** |
> > >
> > > 1. **Training with our Sekai dataset significantly improves camera control accuracy.** We have explored camera-controlled video generation utilizing the camera trajectory annotation in Sekai-Real-Drone. Specifically, given a camera trajectory  , an initial image and a text prompt, Camera-Controlled Video Generation aims to generate a video that follows the specified camera motion while maintaining visual and semantic consistency. Leveraging the annotated precise camera trajectories in Sekai, we compute the Plücker embeddings for each frame and inject them into the model as explicit control signals during fine-tuning.
> > >
> > >     Our experiments were conducted on Sekai-Real-Drone based on Wan2.1-Fun-V1.1-1.3B-Control-Camera. We fine-tune the baseline for 2 epochs, and evaluated on a test set containing 50 randomly selected samples from diverse scenarios. We adopt two metrics from CameraCtrl [1]: TransErr and RotErr, to quantitatively evaluate camera control. These metrics measure the discrepancies in translation and rotation between the input camera trajectory and the trajectory extracted from the generated video using Mega-SAM. Experimental results show that camera control error rates decreased by over 60% on average. This, to some extent, validates the reliability of the pose annotations provided in the Sekai dataset.
> > >
> > > | Method     | TransErr ↓ | RotErr ↓ |
> > > | ---------- | ---------- | -------- |
> > > | baseline   | 28.32      | 27.2     |
> > > | fine-tuned | 7.5        | 9.95     |
> > >
> > > We would greatly appreciate your feedback on whether our responses have resolved your concerns.
> > >
> > > [1] Xiao, Yuxi, et al. "Spatialtrackerv2: 3d point tracking made easy." arxiv. 2025.
> > >
> > > [2] Sturm, Jürgen, et al. "A benchmark for the evaluation of RGB-D SLAM systems." IROS. 2012.
> > >
> > > [3] Rockwell, Chris, et al. "Dynamic Camera Poses and Where to Find Them." CVPR. 2025.
> > >
> > > [4] Mayer, Nikolaus, et al. "A large dataset to train convolutional networks for disparity, optical flow, and scene flow estimation." CVPR. 2016.

---

### Note · Authors · 2025-08-13

Dear Reviewers and Area Chair,

We sincerely thank you all for your thorough reviews, constructive feedback, and engaging discusions throughou the review proces.

**Acknowledged Strengths.** We appreciate the reviewers’ consensus on our key contributions:

- High-quality and first-of-its-kind large-scale first-person world exploration dataset featuring long, dynamic, trajectory-controlled clips with rich annotations, filling a gap and benefiting interactive video generation and other related tasks.
- Large scale and diverse coverage (~6,000 hours from 1,000+ cities, spanning varied locations, weather, crowd density, and viewpoints)
- Rich multi-dimensional annotations (location, scene type, weather, crowd density, captions, camera trajectories) enabling multi-task learning and conditional generation
- Novel and rigorous pipeline (GPU-optimized automated workflow, dual aesthetic–semantic quality filtering, reusable multi-dimensional sampling algorithm) ensuring consistency and reliability

**Rebuttal Efforts Summary.** Through our responses, we have:

- Evaluated video generation models trained on Sekai using larger-scale data, more comprehensively covered test sets, and additional metrics/benchmarks, further validating Sekai's effectiveness.
- Conducted extensive experiments across downstream tasks (camera-controlled and interactive video generation) and diverse data perspectives (drone and game videos), demonstrating Sekai’s versatility.
- Announced improvements to Sekai, including refined category strategies, improved location filtering, and broader camera trajectory annotation coverage, enhancing Sekai's reliability.
- Provided qualitative and quantitative evidence validating the reliability of selected annotation model (e.g. MegaSaM).

**Planned Amendments for Camera-Ready.**

- Integrate extended experimental results together with additional details on experimental settings and result analysis.
- Add annotation quality validation procedures, refined taxonomies, and updated statistics on expanded trajectory coverage.
- Incorporate explicit ethical compliance statement and privacy-preserving strategy.
- open-source the dataset along with the curation pipeline.

We are encouraged that reviewers UV8k, Acq5, and KB8d confirmed their concerns were fully addressed, with KB8d and UV8k raising scores to positive. We look forward to incorporating these improvements in the final version and sincerely thank you again for your valuable contributions to this work.

---

### Decision · Program_Chairs · 2025-09-18

**Decision:**

Accept (poster)

**Comment:**

This paper presents a high-quality, first-person global video dataset with rich annotations for world exploration. The paper initially received mixed reviews, primarily due to insufficient experimental validation and unclear annotation quality. After extensive discussions between the authors and reviewers, all reviewers recognized the paper's contributions and agreed to accept it. AC hopes that all comments will be incorporated into the final version.